# The Role of E3 Ligase Pirh2 in Disease

**DOI:** 10.3390/cells11091515

**Published:** 2022-04-30

**Authors:** Alexandra Daks, Olga Fedorova, Sergey Parfenyev, Ivan Nevzorov, Oleg Shuvalov, Nickolai A. Barlev

**Affiliations:** Institute of Cytology RAS, 194064 St. Petersburg, Russia; fedorovaolga0402@gmail.com (O.F.); gen21eration@gmail.com (S.P.); ban140598@gmail.com (I.N.); oleg8988@mail.ru (O.S.)

**Keywords:** Pirh2, RCHY1, E3 ligase, ubiquitination, cancer, p53 protein family, virus infection, HIV-1, SARS-CoV, nerve injury

## Abstract

The p53-dependent ubiquitin ligase Pirh2 regulates a number of proteins involved in different cancer-associated processes. Targeting the p53 family proteins, Chk2, p27^Kip1^, Twist1 and others, Pirh2 participates in such cellular processes as proliferation, cell cycle regulation, apoptosis and cellular migration. Thus, it is not surprising that Pirh2 takes part in the initiation and progression of different diseases and pathologies including but not limited to cancer. In this review, we aimed to summarize the available data on Pirh2 regulation, its protein targets and its role in various diseases and pathological processes, thus making the Pirh2 protein a promising therapeutic target.

## 1. Introduction

The ubiquitin–proteasome system (UPS) is the main mechanism of non-lysosomal proteolysis that provides protein homeostasis in the cell and ensures the functioning of numerous molecular processes and pathways. Perturbations in the UPS are associated with different types of pathological processes and diseases including cancer, neurodegenerative diseases, viral infections and autoimmune disorders [1,2,3,4]. E3 ligases is a large class of effector enzymes that transfer the ubiquitin mark from the upstream E1 and E2 enzymes to the substrate proteins. Thus, the E3 ligases provide the specificity of ubiquitination reaction either targeting the specific protein for degradation in proteasomes or changing is functional state. To date, E3 ligases are considered as promising therapeutic targets mainly for treating cancers. Particularly, a number of small molecule inhibitors of p53-specific ubiquitin ligase Mdm2 have been developed and some of them are currently in different stages of clinical trials [5,6,7]. Pirh2 was also shown to ubiquitinate the main tumor suppressor protein p53, so it is not surprising that Pirh2 is being studied mostly in the context of p53 regulation. However, Pirh2 was shown to target a number of p53-independent factors thereby regulating such cellular processes as proliferation, cell cycle regulation, apoptosis, and tumor transformation. Accordingly, the participation of Pirh2 in different diseases and pathologies was shown. In this review, we aimed to summarize the evidence reported in the literature about the role of Pirh2 in cancer and other diseases supporting the attractiveness of Pirh2 as a therapeutic target.

## 2. The Structure of Pirh2

An E3 ubiquitin ligase Pirh2 (p53-induced RING-H2 protein) encoded by the RCHY1 gene has a list of alternative names such as CHIMP (CH-rich-interacting match with PLAG1), RNF199 (RING finger protein 199) ZNF363 (Zinc finger protein 363). It was first described as ARNIP (Androgen receptor N-terminal-interacting protein) in 2002 [8]. The RCHY1 human gene is localized on the chromosome 4 in 4p21.1 locus. The gene is coded by 9 exons that produce 261 amino acids that represent the full-length variant of Pirh2.

Together with MDM2, MDM4, BRCA1, TRAF2 and others proteins, Pirh2 belongs to the RING (Really Interesting New Gene) domain-containing family of ubiquitin ligases that comprises more than 600 E3 enzymes according to results of the human genome analysis [9,10].

The structure of Pirh2 is coordinated by nine Zn^2+^ ions, which is provided by high content of cysteine and histidine residues (11% and 8%, accordingly), and includes a central RING domain providing ubiquitin ligase activity and amino- and carboxy-terminal domains that are mostly involved in protein–protein interactions of Pirh2 [11]. (Figure 1).

The RING domain of Pirh2 is composed by 138–189 amino acid residues and contains the RING-H2 motif (Cys_3_His_2_Cys_3_) to which two Zn^2+^ ions are chelated [11]. As was mentioned above, this domain is responsible for enzymatic activity of Pirh2, and mutations at M176 and C186 positions lead to a loss of ubiquitin ligase activity [11]. The amino-terminal domain of Pirh2 (NTD) that contains the zinc finger motif and the carboxy-terminal domain (CTD) provides an interaction of Pirh2 with other proteins. The deletion mutant of Pirh2 lacking the NTD is able to bind several target proteins and ubiquitinate them, while the point mutant lacking the CTD retains the ability of self-ubiquitination [11].

To date, five isoforms of the Pirh2 proteins have been reported: Pirh2A (full-length isoform), B, b, C and D. The first described Pirh2 isoforms were Pirh2B and Pirh2C [12]. These two isoforms are expressed in different cell types and organisms as a result of the alternative splicing. Thus, mRNA of the Pirh2B isoform lacks the exon 7 encoding amino acids 171–179. In the Pirh2C isoform, in turn, 180–186 amino acids of the RING domain and the entire CTD are lacking [12] (Figure 1). The expression of Pirh2B and Pirh2C was shown in different types of cancer cell lines including lung cancer H1299 cells, colon cancer cells RKO and HCT116. Surprisingly, it was demonstrated that these isoforms are also able to downregulate the protein levels of Pirh2 targets [12].

Analysis of cDNA libraries of human liver cells revealed another Pirh2 isoform—Pirh2b [13]. This Pirh2 protein variant is formed as a result of alternative splicing which results in deletion of 38 nucleotides in the 8th exon and a premature stop codon. As a result, Pirh2b shares the 1–179 amino acid sequence with Pirh2C isoform and additionally has nine unique amino acids in the C-terminal region [13] (Figure 1). It was shown that Pirh2b is still able to interact with targets of the full-length Pirh2 form but is not able to ubiquitinate them. Of note, the expression level of Pirh2b in hepatocellular carcinoma cells is lower than in non-cancer liver cells suggesting that cancer cells lack this compensatory mechanism [13].

The smallest known Pirh2 isoform—Pirh2D—was discovered by analyzing the EST (expressed sequence tags) database. Pirh2D consists of 75 amino acid residues and is formed as a result of the adenine base insertion, which generates a reading frame shift and the unmature stop codon formation. As a result, Pirh2D includes first 67 amino acids that match all other isoforms and an additional 8 unique amino acids [14] (Figure 1). To date, the role and tissue-specific distribution of this isoform in is not clear.

Taking into account the well-established role of Pirh2 in the development and progression of various pathological processes which will be described below, we believe that detailed investigation of Pirh2 isoforms and their participation in the development of diseases has great prospects in the frame of biomedical research.

## 3. Pirh2 Regulation

The first transcription factor described to regulate the Pirh2 expression was a tumor suppressor p53. The p53-binding site in intron 3 of the RCHY1 mouse gene was identified and p53-dependent activation of RCHY1 in mouse erythroleukemia cells was shown [15] (Table 1). Surprisingly, in human cancer cell lines the Pirh2 level was shown to be independent of the wild-type p53 protein, which is in drastic contrast to the Mdm2 gene, which is regulated by p53 both in mice and humans [16].

Later, our group showed that the other member of the p53 protein family—p63—acts as an activator of RCHY1 expression in human osteosarcoma cells [17]. Importantly, we demonstrated that full-length form of p63 (TA-p63) but not the isoform lacking the transactivating domain (ΔN-p63), positively regulates Pirh2 transcription (Table 1). Since all three members of the p53 family—p53, p63 and p73—are known to share a large number of common target genes due to the high structural homology and the ability to bind common response elements, we and others hypothesized that p73 potentially can also act as a transcription regulator of RCHY1 [26]. However, this hypothesis requires further experimental confirmation.

A recent study demonstrated the ability of RelA/p65 subunit of NFkB bind to the promoter of Pirh2-coding gene and activate its expression [18]. Additionally, the negative effect of Naa10p acetyltransferase (N-α-Acetyltransferase 10 protein) on RelA/p65 phosphorylation and its ability to activate Pirh2 expression was shown [18] (Table 1).

We performed a bioinformatic analysis of the Pirh2-coding gene sequence to reveal potential transcription factors regulating Pirh2 expression using the SEA MEME Suite [27]. Besides p53 itself, we revealed the potential binding sites for such transcription factors as ZN121, ZN770, MEF2D, PITX2, MAZ, VSX2, HEN1, EGR2, PAX2, TBX2 and TBX15 (Figure 2). Undoubtedly, the ability of these factors to regulate Pirh2 needs the experimental confirmation and detailed analysis in the future.

Interestingly, Pirh2 itself was shown to undergo a list of posttranslational modifications such as ubiquitination, phosphorylation and SUMOylation. However, only a handful of enzymes are known to date that covalently modify Pirh2.

One of the known Pirh2-specific targets is the CamKII enzyme (Ca^2+^/calmodulin-dependent protein kinase II). CamKII is a multifunctional serine/threonine kinase targeting various proteins participating in neurotransmission, cytoskeleton functioning, learning and memory, synaptic plasticity and gene expression regulation such as AMPA-R [28,29], filamin A [30], β-catenin [31], CREB [32], and others. Dysregulation of CamKII is associated with different pathological conditions including neurodegenerative, cardiovascular, metabolic and malignant diseases, and inhibition of CamKII is considered as a promising therapeutic strategy for the heart failure, arrhythmia and diabetic heart dysfunction [33,34]. CamKII was shown to phosphorylate Pirh2 at T154 and S155 augmenting its auto-ubiquitination and consequently stabilizing its target—p53 [19] (Table 1). It was demonstrated that, while in normal tissues Pirh2 mostly exists in the phosphorylated form, Pirh2 is mainly unphosphorylated in cancer cells and primary tumor samples. In line with this notion, CamKII-mediated phosphorylation of Pirh2 was shown to inhibit tumor growth in xenograft models [19].

Cyclin-dependent kinase Cdk9 is one of the key general regulators of transcription through phosphorylation of RNAP II (RNA polymerase II). Mechanistically, Cdk9 forms the complex with cyclin T1, and this complex named Positive Transcription Elongation Factor b (P-TEFb) carries out phosphorylation of RNAP II [35,36]. It was repeatedly shown that Cdk9 promotes proliferation, survival and metastasis of cancer cells and high Cdk9 expression is associated with unfavorable outcome of different types of cancers [37,38,39,40]. Several small-molecule inhibitors of Cdk9 were developed and some of them reached different stages of clinical trials [41,42,43]. Cdk9 was shown to also phosphorylate Pirh2 protein at S211 and T217 [20] (Table 1). Similar to CamKII, phosphorylation of Pirh2 by Cdk9 stimulates autoubiquitination activity of Pirh2 and leads to p53 stabilization in glioblastoma cells.

Another factor involved in Pirh2 downregulation on the protein level was shown to be the Homeodomain Transcription Factor Hoxa2—a homeobox protein playing an important role in embryogenesis and involved in a number of pathologies including cancer [44,45,46,47]. Hoxa2 acts mainly as a transcription regulator binding to regulatory elements of different genes, but also may affect the proteins functions, stability and cellular distribution through direct interactions with its partners [47,48]. Thus, it was demonstrated that Hoxa2 physically interacts with Pirh2 and 20S proteasome subunits facilitating the proteasomal degradation of Pirh2 in the ubiquitin-independent manner [21] (Table 1). Functionally, Hoxa2-dependent degradation of Pirh2 led to p53 stabilization. Further investigations of this research group revealed that both 19S and 20S proteasome subunits are involved in Hox2-dependent Pirh2 degradation [22]. Interestingly, this regulatory mechanism is evolutionarily conserved among different vertebrate species [22]. Since it has been shown that the 20S proteasomal complex can associate with microRNA [49] and Hoxa2 can also bind nucleic acids, it would be interesting to see whether this interaction is at least partially regulated by microRNAs.

One of the known positive Pirh2 regulators is histone acetyltransferase KAT5, or Tip60 (Tat-interactive protein of 60 kDa). Being a multifunctional factor, in addition to acetylation of histones H2, H3 and H4, Tip60 also acetylates such proteins as p53, ATM, AR and others, and thus participates in crucial cellular processes including proliferation, cell cycle regulation, DNA damage response and apoptosis [50,51,52,53,54]. Considering a wide range of Tip60 targets, different cancer types such as prostate cancer, melanoma, colorectal cancer and others are expectedly characterized by altered Tip60 expression [55,56,57,58]. Tip60 was shown to interact with and stabilize Pirh2 [23] (Table 1). Moreover, it was demonstrated that Tip60 overexpression led to Pirh2 translocation to the nucleolus where partial co-localization of Pirh2 and Tip60 was observed [23]. The effect of Tip60-dependent re-localization of Pirh2 was more pronounced in human colon cancer cells (PC3M and LNCaP) cultured in steroid-depleted medium. However, the functional significance of this interaction remains to be elucidated.

Furthermore, the PLAGL2 (pleomorphic adenoma gene like-2) transcription factor is also involved in the regulation of post-translational modifications of Pirh2. PLAGL2 is a member of PLAG zinc-finger transcription factors family. Enhanced expression of PLAGL2 was shown to promote lung cancer, colorectal cancer, hepatocellular carcinoma (HCC), and many other cancer types [59,60,61]. Proto-oncogene PLAGL2 was shown to promote epithelial–mesenchymal transition (EMT) and colorectal metastasis through β-catenin-dependent Zeb1 activation [62]. PLAGL2 also activates the expression of USP34 deubiqutinase that in turn stabilizes Snail1 and promotes proliferation and migration of gastric cancer cells [63]. In HCC cells, PLAGL2 expression was shown to be regulated by Hif1a via a direct binding of Hif1a to the PLAGL2 promoter [60]. Another regulatory mechanism involves Hif1a-dependent activation of lncRNA MAPKAPK5-AS1 expression that sponges PLAGL2-specific microRNA miR-154-5p [64]. PLAGL2 in turn, activates EGFR expression thereby activating PI3K/AKT signaling pathway. The latter pathway positively regulates Hif1a, thus forming a positive feedback loop [60,64]. In 2007, Zheng et al. demonstrated that PLAGL2 physically interacted with Pirh2 dimers and this interaction stabilized Pirh2 on the protein level preventing its degradation in proteasomes [24] (Table 1). The authors hypothesized that PLAGL2 being an oncogene may carry out indirect downregulation of p53 in cancer cells through Pirh2 stabilization and thus promote the progression of carcinogenesis. However, this speculation needs further experimental confirmation.

To date, there are five ubiquitination sites and one SUMOylation site which have been revealed in the Pirh2 amino acid sequence using mass spectrometry [65,66,67,68,69]; however, there is only one E3 ligase known to ubiquitinate Pirh2. Thus, Yang et al. demonstrated that RNF144B interacts with Pirh2, and targets it for degradation through the ubiquitin–proteasome system in gastric cancer cells [25] (Table 1). It was shown that RNF144B-mediated destabilization of Pirh2 in turn led to p53 upregulation and that the miR-100-RNF144B-Pirh2-p53 pathway is involved in apoptosis regulation [25].

To date, there are no data on SUMO ligases performing Pirh2 modification. Additionally, despite Pirh2 autoubiquitination was repeatedly shown [19,20], the sites of this self-modification are still not defined.

Thus, the mechanisms of Pirh2 regulation are still poorly understood and need further investigation. In our recent proteomic study, we analyzed the interactome of Pirh2 and revealed more than 200 novel Pirh2-interacting partners [70]. There are several enzymes performing post-translational protein modifications such as SUMO ligase RanBP2, deubiquitinase USP9X, Poly (ADP-ribose) polymerase PARP1, E3 ubiquitin ligase Rad18 and others. These proteins are potential Pirh2 regulators, whose role in orchestration of Pirh2 functioning, localization and stability requires in-depth research.

## 4. The Protein Targets of Pirh2

### 4.1. p53 Family Proteins

The p53 tumor suppressor protein is a key regulator of cellular response to such stressful stimuli as DNA damage, oncogene activation, and hypoxia. In response to stress, p53 activates many transcriptional targets involved in cell cycle arrest, DNA damage repair and initiation of apoptosis: p21, Puma, Bax, Noxa, and others [71]. The amount and activity of p53 in the cell is strictly regulated by post-translational modifications such as methylation, acetylation, phosphorylation, and ubiquitination [72,73].

The p53 protein level is kept low in cells under normal conditions which is achieved mainly through the ubiquitin-dependent proteasomal degradation [74]. To date, more than 20 p53-modifying ubiquitin ligases have been discovered including Mdm2, COP1, ARF-BP1, synoviolin, E4F1, WWP1 as well as Pirh2 (reviewed in [75]). It is now generally accepted that Mdm2 is the main negative regulator of p53 [76,77,78], and a number of Mdm2 inhibitors display various degrees of promise as anticancer agents aimed to restore p53 functions [79].

Pirh2 was repeatedly shown to polyubiquitinate p53 on multiple lysines leading to its subsequent degradation in proteasomes [11,15,80]. Interestingly, it was demonstrated that Pirh2 preferentially binds to tetrameric form of p53 possessing transcriptional activity [11]. Additionally, in contrast to Mdm2, Pirh2 is able to ubiquitinate p53 phosphorylated at the N-terminally located S15 [80]. This modification mediated by ATM kinase occurs in response to DNA damage [81] and prevents the Mdm2–p53 binding [82]. Importantly, mice with knockout of the *RCHY1* gene encoding Pirh2 are viable, in contrast to Mdm2 knockout, which are lethal [83,84]. Accordingly, ionizing radiation of cells from the Pirh2^-/-^ mice led to a significant increase in the number of p-p53(Ser15) followed by the augmentation of expression levels of its target genes and the level of apoptosis, compared to cells from the wild-type mice [84]. As a result, Pirh2 knockout cells were shown to be more sensitive to ionizing radiation due to increased p53 activity [84]. These data indicate the key role of Pirh2 in p53 regulation under conditions of genotoxic stress.

Interestingly, Pirh2 promotes proteasomal degradation of not only the wild-type, but also the mutant form of p53. Thus, it was shown that arsenic trioxide being an anticancer drug used in clinical practice stabilizes Pirh2 and promotes proteasome-dependent degradation of mutant forms of p53 [85]. In response to stress, Pirh2 interacts with, ubiquitinates and downregulates two mutant forms of p53, R248W and H179Y/R282W, which are abundantly expressed in MiaPaCa-2 and HaCaT human cell lines [85,86]. It should be noticed here that Mdm2 is also able to downregulate the basal level of mutant p53 [87,88] although to a much lesser extent compared to the wild-type form of p53 [89].

Besides p53 itself, the p53 protein family includes two other transcription factors—p63 and p73. All three p53 family members share high structural and functional homology. Similar to p53, p63 and p73 proteins are also tumor suppressor transcription factors that activate pro-apoptotic genes and cell cycle regulators in response to genotoxic and other forms of stress [90]. Due to the high degree of homology between transactivating domains (TA domains) of p53, p63 and p73 these proteins bind the same p53 response elements (p53 RE) of target genes to regulate their expression [90]. Besides cellular functions common between all three proteins, p63 and p73 play roles in embryogenesis that differ from the one played by p53. Specifically, p63 is involved in the formation of limbs and all epithelial organs, such as skin, mammary and salivary glands [91]. In turn, p73 is one of the key regulators of the nervous system development and lack of p73 leads to underdevelopment of the hippocampus and the absence of several types of neurons of the central and peripheral nervous system [92]. Interestingly, the developmental functions of p63 and p73 are performed mostly by their truncated ΔN isoforms lacking the TA domains [93].

In this respect, p63 was shown to be a target of Pirh2-dependent ubiquitination in human keratinocytes. Importantly, Pirh2 was able to ubiquitinate both the full-length p63 protein and its ΔN isoform targeting them for proteasomal degradation [94]. ΔN-p63 promotes the proliferation of undifferentiated keratinocytes, while Pirh2-dependent downregulation of ΔN-p63 in HaCaT cells contributes to keratinocyte differentiation [94].

Similar to p53 and p63, p73 was shown to be ubiquitinated by Pirh2 [95,96]. The research group of Dr. Chen demonstrated that Pirh2 polyubiquitinates the full-length TA isoform of p73 with its subsequent degradation by 20S and 26S proteasomes [96]. Further, it was shown that knock-down of Pirh2 led to an increase in p73-induced apoptosis in response to doxorubicin treatment in H1299 cells, while Pirh2 overexpression has the opposite effect [96]. Surprisingly, another study performed using HCT116 p53^-/-^ cell line reported somewhat contradictory effects of Pirh2-mediated ubiquitination of p73. Thus, according to the report of Wu et al., Pirh2 indeed ubiquitinated p73, but this modification did not induce p73 degradation. Instead, ubiquitination of p73 by Pirh2 significantly decreased transcriptional activity of p73 and subsequent p73-induced cell cycle arrest, but not apoptosis [95]. The follow-up study from the same research group found an additional mechanism of Pirh2-mediated regulation of p73. The HECT domain-containing E3 ligase, AIP4, which promotes p73 ubiquitination and degradation, was shown to be downregulated by Pirh2 on the protein level. As a result, Pirh2 contributed to p73 stabilization and enhanced its tumor suppressor activity, resulting in G1-specific cell cycle arrest [97]. Thus, Pirh2, being found to be a common regulator of all three members of the p53 family, is likely to be one of the major as yet underappreciated regulators of cell cycle progression, DNA repair, apoptosis, embryogenesis and tumor transformation.

### 4.2. c-Myc Oncogene

Proto-oncogene c-Myc, a product of the *MYC* gene, is a transcription factor that is upregulated in approximately half of tumors [98]. By binding to promoter regions of its target genes c-Myc regulates numerous cancer-associated cellular processes including proliferation, apoptosis, metabolism, metastasis, epithelial-mesenchymal transition and stemness [70,99,100,101,102].

It was reported that Pirh2 mediates polyubiquitination of the c-Myc protein targeting it for proteasomal degradation [84]. Pirh2 mutant mice used in that study demonstrated an increased level of spontaneous tumors formation, caused by elevated c-Myc levels in different tissues [84]. According to this study, the effect of Pirh2-induced downregulation of c-Myc level was not limited by mouse models but also manifested in RKO human colon carcinoma cells [84]. Interestingly, thymocytes and splenocytes isolated from Pirh2 deficient mice demonstrated increased apoptosis compared to Pirh2 wild-type cells in response to irradiation [84].

On the other hand, the studies from our group revealed the stabilizing effect of Pirh2 on c-Myc both at the mRNA and protein levels [70,103]. We revealed the oncogenic properties of Pirh2 in human non-small cell lung carcinoma cells (NSCLC) H1299 and suggested that they can be mediated by Pirh2-induced up-regulation of c-Myc [103]. Later, we demonstrated the ability of Pirh2 positively regulate c-Myc in another human cancer cell line—HeLa [70]. We managed to identify the molecular mechanism of this regulation, which will be discussed in the next subsection.

### 4.3. RNA Binding Protein HuR

The RNA binding protein HuR (human antigen R) is a product of *ELAVL1* (embryonic lethal abnormal vision 1) gene and a member of the ELAV/Hu protein family. In the nucleus, HuR participates in splicing and transport of mRNAs through the nuclear pore complex [104,105]. In the cytoplasm, HuR was shown to bind 3′- or 5′-UTR regions of its several target mRNAs affecting their stability both positively and negatively [106]. In this respect, HuR was shown to stabilize MMP9 [107], cyclins A2 and B1 [108] and Hif1a [109] mRNAs among others. On the other hand, HuR was also shown to repress several mRNAs including p27 [110], RhoB [111], and c-Myc [112]. Due to a broad range of HuR-targeted mRNAs whose products are involved in carcinogenesis, HuR itself participates in many aspects of tumorigenesis and is considered as a biomarker and an important potential target for anticancer therapy [106,113].

We have demonstrated that Pirh2 physically interacts with and polyubiquitinates the HuR protein [70]. According to our data, Pirh2-mediated ubiquitination targets HuR for proteasomal degradation and this result was confirmed by different studies in multiple cell types [70]. Importantly, HuR was shown to promote the degradation of c-Myc mRNA by recruiting the RISC complex [112]. In line with this finding, we demonstrated that Pirh2 suppressed HuR at the protein level that, in turn, led to stabilization of c-Myc mRNA and subsequently the protein accumulation [70]. Thus, we revealed an alternative mechanism of Pirh2-mediated regulation of c-Myc that involves the RNA binding protein, HuR [70]. It is known that under normal conditions the HuR protein demonstrates nucleoplasmic localization, while in response to different stress stimuli such as heat shock, oxidative stress, or DNA damage, it relocates into the cytoplasm and becomes a part of stress granules [114,115]. Importantly, HuR is absent from the nucleoli both under normal and stress conditions [116]. Taking into account that Pirh2 is localized both in the nucleus and cytoplasm and the ability of Pirh2 to be recruited to nucleolus [23], we hypothesize that Pirh2 participates in regulation of HuR subcellular localization.

### 4.4. DNA Polymerase Eta

Along with Pol ι, Pol κ, REV1, Pol ζ, Pol μ, Pol λ, Pol ν, and Pol θ, DNA polymerase Eta (Pol η) is involved in trans-lesion DNA synthesis. Pol η is able to bypass UV-induced cyclobutane pyrimidine dimers (CPDs), platinum-induced intrastrand crosslinks (Pt-GG), and 6-4 photoproducts [117,118,119]. Being bulky lesions, CPDs are the major cause of distortion of the DNA structure and replication process failure. Pol η is one of the key enzymes that helps repair these lesions [120]. However, being an error-prone polymerase that lacks the proofreading activity, Pol η causes the formation of a large number of mutations as a result of its activity [121,122,123]. Loss of the Pol η function causes an autosomal recessive disease xeroderma pigmentosum variant (XPV) which is manifested by hypersensitivity to UV irradiation and cancer predisposition [124]. On the contrary, increased Pol η expression promoted chemotherapeutic drug-induced mutations as well as resistance to therapy [119,125,126].

Jung et al. demonstrated that Pirh2 interacts with and downregulates Pol η, directing it for degradation in the 20S proteasomes in a ubiquitin-independent manner [123]. Additionally, it was shown that Pirh2 knock-down led to accumulation of Pol η and increased UV resistance of cancer cells. Later, the same group uncovered that Pirh2 performed monoubiquitination of Pol η that leads to dissociation of the Pol η–PCNA complex that results in suppression of trans-lesion synthesis aimed to overcome the UV-induced DNA damage [127]. Thus, according to these studies, Pirh2 prevented the bypass of UV-induced lesions and thus sensitized H1299 and RKO human cancer cells to UV damage. Mechanistically, Pirh2-mediated monoubiquitination of Pol η led to inhibition of its ability to interact with PCNA concomitantly promoting degradation of Pol η in 20S proteasomes [127,128].

Notably, the controversial data regarding the role of Pirh2 in cellular susceptibility to UV treatment were reported by Duan et al. [129]. Pirh2 was shown to interact with keratin 8/18 (K8/18) without affecting its stability. According to this study, Pirh2–K8/18 interaction was crucial for K8/18 filament organization and mitochondrial distribution. Disruption of the Pirh2–K8/18 protein complex via both Pirh2 suppression or UV treatment led to abnormal mitochondrial aggregation [129]. The authors also demonstrated that knockdown of Pirh2 significantly increased the level of apoptosis in H1299 cell subjected to UV treatment [129]. Thus, the role of Pirh2 in cellular response to DNA damage induced by UV irradiation needs further investigation.

### 4.5. The Chk2 Protein Kinase

Another critical protein involved in DNA damage response (DDR) is a serine/threonine protein kinase Chk2. This Chk2 kinase itself is phosphorylated by ATM kinase in response to double-stranded DNA lesions [130]. Upon this activating phosphorylation by ATM, Chk2 phosphorylates numerous downstream targets involved in DDR such as p53 [131], E2F1 [132], XRCC1 [133], BRCA1 [134] and many others, providing an appropriate cellular response to stress stimulus. Importantly, Chk2 controls G1-S, S and G2-M checkpoints ensuring cell cycle arrest for effective DNA damage repair [135,136,137]. Numerous studies indicate that Chk2 is a tumor suppressor protein whose dysfunction is associated with various types of cancer [138,139,140,141]. On the other hand, inhibitors of checkpoint kinases including Chk2 are considered to be promising therapy agents aimed to prevent DNA repair in cancer cells [142,143,144,145,146].

It was shown that the cells of Pirh2^−/−^ mice are characterized by elevated Chk2 expression level both under normal conditions and genotoxic stress [147]. Mechanistically, Pirh2 polyubiquitinates Chk2 and targets it for degradation. Importantly, phosphorylation of Ser460 of murine Chk2 (equivalent to human Ser456) prevents Pirh2-mediated Chk2 degradation, while the mutant form of Chk2 which cannot be phosphorylated at this site demonstrated Pirh2-induced hyper-ubiquitination and increased proteasomal degradation [147]. Since Ser456 phosphorylation is considered to be a stabilizing covalent modification of Chk2 [148], one could hypothesize that Pirh2 affects Chk2 turnover mostly under normal conditions. Indeed, Pirh2 knockout reinforced Chk2-mediated activation of G1/S and G2/M cell-cycle checkpoints [147].

### 4.6. The Cyclin-Dependent Kinase (CDK) Inhibitor p27

The cyclin-dependent kinase (CDK) inhibitor p27^Kip1^ (hereinafter p27) acts as regulator of cell cycle progression at different stages. The major and first identified p27 target is CDK2 that forms the complex with cyclin E in G1 phase and promotes G1/S transition [141]. In the early S phase, CDK2 binds to cyclin A and provides DNA replication [149,150]. The maximum level of p27 is observed in G0/G1 phases and then drops sharply before the S phase due to CDK2-induced phosphorylation of p27 on T187. This modification promotes p27 ubiquitination by ubiquitin ligase Skp2 and subsequent degradation in proteasomes, which allows the cell to enter the S phase [151,152,153]. Additionally, p27 was shown to downregulate CDK4/6 and CDK1 and thus participate in regulation of corresponding cell cycle phases [154,155,156]. The level of p27 is elevated in response to DNA damage that provides cell cycle arrest for effective DNA repair [157,158]. The expression of p27 is downregulated in different cancer types including lung cancer, breast cancer and colon cancer, and associated with poor prognosis [159,160,161].

Pirh2 acts as ubiquitin ligase for p27 and promotes its polyubiquitination and proteasomal degradation [162]. It was demonstrated that Pirh2 expression is induced from late G1 towards the S phase and Pirh2 contributes to p27 degradation and cell cycle progression at G1-S transition in human glioblastoma cells [162]. Interestingly, Pirh2 knock-down has more significant effect of blocking cell cycle progression compared to two another known p27-specific E3 ligases—Skp2 and KPC1, which testifies to the crucial role of Pirh2 in regulation of cell proliferation through downregulation of p27.

### 4.7. Histone Deacetylases HDAC1 and HDAC2

Class I histone deacetylases HDAC1 and HDAC2 are highly homologous enzymes (about 83% identity) performing deacetylation of histones H2A, H2B, H3 and H4 which results in chromatin tightening and silencing of gene expression [163]. Both HDAC1 and HDAC2 are involved in regulation of cell cycle, cellular senescence and cancer progression [164]. To date, histone deacetylases are considered to be promising therapeutic targets and a number of HDAC1 and HDAC2 inhibitors are tested as anticancer agents at different stages of clinical trials [165,166].

As was mentioned above, Pirh2 was first described in 2002 as androgen receptor (AR) N-terminal-interacting protein ARNIP [8]. Later, in 2006, Logan et al. confirmed Pirh2–AR interaction and demonstrated that Pirh2 is recruited to the ARE (androgen response element) of the PSA (prostate-specific antigen) coding gene enhancing AR-mediated PSA expression activation in human prostate adenocarcinoma cells [167]. Importantly, in this study Pirh2-mediated polyubiquitination and proteasomal degradation of deacetylase HDAC1 [167]. Since HDAC1 is one of the key negative regulators of AR expression, the dual positive regulatory effect of Pirh2 on AR was demonstrated.

Recent study performed by Choi et al. revealed the ability of Pirh2 to suppress HDAC2 [168]. The negative effect of Pirh2 on both HDAC1 and HDAC2 levels was demonstrated in several human cancer cell lines including MCF7, HCT116, H1299 and others which apparently indicates the universality of Pirh2-dependent regulation of these histone deacetylases [168]. Interestingly, valproic acid (VPA) being a selective inhibitor of HDAC class I positively regulates the Pirh2 level [168], which indicates the possibility of mutual negative regulation of Pirh2 and HDAC1/2.

Thus, Pirh2 may be considered as not only the key AR regulator but also as important participant of gene expression machinery.

### 4.8. Twist1

Twist1 is one of the key transcription factors that regulate the EMT process during embryogenesis, tissue regeneration and metastasis formation. Twist1 was shown to downregulate such epithelial markers as E-cadherin, cytokeratins and estrogen receptor (ER) and, conversely, upregulate mesenchymal factors including Vimentin, N-cadherin, metalloproteases and integrins [169,170]. Mutations in the corresponding gene are associated with Saethre–Chotzen syndrome, which is an autosomal dominant disease manifested by coronal synostosis and mild limb deformities [171]. Twist1 overexpression was shown to trigger tumor transformation, stemness, metastasis, and resistance to chemotherapy in a variety types of cancer [169,172].

Rather surprisingly, an oncogenic E3 ligase, Pirh2, was shown to polyubiquitinate Twist1 and target the latter for proteasomal degradation [173]. Furthermore, p53 turned out to be an important mediator of the Pirh2–Twist1 association that leads to degradation of Twist1 [173]. Notably, hot-spot mutations in the p53 (R175H, R248W, R273H) abrogate its ability to promote Twist1 degradation leading to activation of EMT and formation of an invasive phenotype [173].

### 4.9. SCYL1-BP1

The SCYL1-binding protein (BP1) was identified as a Pirh2-interacting protein in 2005 [174]. To date, functions of SCYL1-BP1 are not well characterized; however, it is known that mutations in the SCYL1-BP1-coding gene are associated with such premature aging diseases as *geroderma osteodysplasticum* [175]. Additionally, SCYL1-BP1 was shown to participate in cell cycle regulation via stabilization of Cyclin F and RRM2 [176]. Importantly, it was also reported that SCYL1-BP1 augmented p53 functions through promoting auto-ubiquitination of MDM2 and hence preventing the MDM2-mediated degradation of p53 [177]. According to this study, SCYL1-BP1-mediated stabilization of p53 enhanced the transcriptional activity of the latter, increased apoptosis and suppression of tumorigenicity of human hepatic adenocarcinoma cells both in vitro and in vivo [177]. In turn, Pirh2 was shown to ubiquitinate the SCYL1-BP1 protein and target it for proteasomal degradation [178]. Thus, inhibition of the p53 positive regulator, SCYL1-BP1, may be another mechanism by which Pirh2 represses p53.

In this section, we try to summarize Pirh2 substrates known to date to demonstrate the diversity of its targets that are involved in various pathological processes. The published data on biological effects of Pirh2 in various diseases are discussed below.

## 5. The Role of Pirh2 in Different Types of Cancer

### 5.1. Lung Cancer

Lung cancer is one of the most frequent causes of worldwide mortality from cancer. According to the National Cancer Institute (NIH NCI) statistics, the 5-year survival rate after the initial diagnosis with lung cancer does not exceed 25%. Several studies demonstrated that Pirh2 is involved in lung cancer progression. Duan et al. carried out the analysis of Pirh2 expression in human lung neoplasms paired with normal lung tissues samples and demonstrated that Pirh2 expression was increased in 84% of lung cancer human specimens [179]. This research group has also shown that Pirh2 overexpression in lung cancer samples correlated with enhanced p53 ubiquitination and degradation, thus contributing to the oncogenic effect of Pirh2 in lung cancer tissues [179]. The further study from that group demonstrated that the level of Pirh2 is elevated in human non-small lung carcinoma (NSCLC) samples compared to normal lung tissues [16] (Table 2).

Another study demonstrated a similar proportion (79.2%) of Pirh2-positive specimens isolated from patients with lung cancer [180]. Moreover, Su et al. reported that knockdown of Pirh2 in A549 NSCLC cell line significantly decreased tumor growth in xenograft rat models [180] (Table 2). These authors also tested the effect of Pirh2 knockdown on A549 cells and showed that the ablation of Pirh2 expression led to reduced cell proliferation, cell cycle block at G0/G1 phase and increased apoptosis [181].

The data on the role of Pirh2 in lung cancer progression obtained by our research group are in agreement with investigations described above. We demonstrated that Pirh2 enhances the tumorigenic phenotype of NSCLC cells [103] (Table 2). Specifically, Pirh2 overexpression increased cell proliferation, resistance to doxorubicin, migration potential of H1299 cells, while ablation of Pirh2 reversed these phenotypes. Since H1299 cells lack p53, we inferred that the tumorigenic effect of Pirh2 manifests p53 independently. Later, as was described above, we demonstrated that Pirh2 positively regulates the c-Myc protein by promoting the degradation of its negative regulator, HuR, which is often overexpressed in lung cancers. Our results may also explain the oncogenic properties of Pirh2 in lung cancer cells [70]. In the follow-up study, Pirh2 was shown to be involved in the autophagy regulation. Knockdown of Pirh2 in different human lung cancer cell lines decreased the expression of genes involved in all steps of autophagy and suppressed the overall autophagy level, leading to sensitization of cells to doxorubicin [182] (Table 2).

### 5.2. Prostate Cancer

The AR signaling is one of the critical pathways involved in the prostate cancer formation and progression. As discussed above, Pirh2 was shown to regulate the AR activity in prostate cancer cells both directly by recruiting the AR transcription factor to the ARE sequence located in the promoter region of the PSA-coding gene and, indirectly, by downregulating the chromatin repressor, histone deacetylase HDAC1 [167]. Consistently, ablation of Pirh2 led to suppression of the prostate cancer cells’ proliferation. Indeed, Pirh2 being expressed in 89% of tumor samples was demonstrated to be a negative prognostic factor for prostate cancer, and its high expression strongly correlates with the aggressiveness of the disease [167] (Table 2).

### 5.3. Oral Cancer

In the study published by Zheng et al., a high level of Pirh2 expression negatively correlated with overall survival and relapse-free survival of patients with oral squamous cell carcinoma (OSCC) [183] (Table 2). Later, the same research group demonstrated that Naa10p suppressed Pirh2 level through inhibiting the NFkB (RelA/p65)-dependent transcription of the corresponding RCHY1 gene. Suppression of Pirh2 positively regulated the p53 signaling in OSCC cells [18]. In addition, Pirh2 knockdown led to suppression of the metastatic activity in OSCC cells through the attenuation of metalloprotease MMP2 expression. The effect of Pirh2 knockdown was partially ameliorated by Naa10p ablation [18].

### 5.4. Hepatocellular Carcinoma

Pirh2 was repeatedly shown to be a prognostic marker for hepatocellular carcinoma (HCC) since both mRNA and protein levels of Pirh2 were elevated in HCC tissue samples compared to both normal or pre-malignant paracarcinomatous liver tissues [184]. High Pirh2 expression strongly correlated with aggressiveness of the disease and unfavorable prognosis for HCC patients [184] (Table 2). These findings were confirmed by another study, according to which Pirh2 expression correlated significantly with histologic grade, venous invasion, tumor size, the presence of multiple tumor-bearing lymph nodes and shortened the lifespan of patients with HCC [185]. This study also demonstrated that Pirh2 and p27 levels correlated inversely and the p27 expression had the opposite impact on HCC aggressiveness and prognosis (Table 2). In support of this observation, in vitro experiments confirmed that Pirh2 can in fact downregulate the p27 protein expression in HCC cells [185].

### 5.5. Head and Neck Cancer

The inverse correlation between Pirh2 and p27 levels as well as their opposite prognostic values for head and neck squamous cell carcinoma (HNSCC) was also reported [186] (Table 2). Knockdown of Pirh2 led to stabilization of the p27 protein resulting in an increase in doubling time of HNSCC cell lines Ho-u-1 and Ho-1-N-1. As in the case of HCC, increased Pirh2 level of expression characterized the higher-grade HNSCC and was an unfavorable prognostic marker for overall survival of HNSCC patients [186].

### 5.6. Glioma

Glioma is the most common primary malignant tumor of the central nervous system in adults that has an unfavorable prognosis in spite of the progress in neurosurgery, as well as in chemo- and radiotherapies. The overall 5-year survival of patients with glioma does not exceed 30–35% (NIH NCI statistics; [191]). It was revealed that Pirh2 protein level correlated with the glioma grade and Pirh2 expression was associated with increased malignancy and poor prognosis in glioma patients [187]. Attenuation of Pirh2 expression promoted apoptosis and inhibited proliferation of U87MG human glioma cells in vitro [187] (Table 2).

The controversial data on the role of Pirh2 in glioma cells were obtained using another human glioma cell line, U251 [188]. It was shown that Pirh2 knockdown reduces apoptosis level in U251 in response to irradiation [188] (Table 2). The authors suggested that this effect was due to the ability of Pirh2 to downregulate an anti-apoptotic factor YAP, but these findings apparently need additional confirmation.

### 5.7. Breast Cancer

The oncogenic role of Pirh2 in breast cancer (BC) was also demonstrated. Pirh2 up-regulation was shown in BC samples compared to tumor-adjacent tissues [189] (Table 2). High Pirh2 expression was highly associated with tumor size and grade, as well as with ER- and Ki-67-positive staining. Moreover, in vitro analysis of Pirh2 knockdown effect in a MDA-MB-231 triple negative BC cell line revealed reduced proliferation and increased G0/G1 cell count. Additionally, ablation of Pirh2 led to downregulation of the well-known oncogene, β-catenin, which is a pro-proliferative transcription factor involved in the Wnt signaling. According to this, Pirh2 was shown to be an unfavorable prognostic marker for BC patients [189].

### 5.8. Multiple Myeloma

Proteasome inhibitor bortezomib is considered to be one of the most effective drugs for treatment multiple myeloma (MM) that accounts for approximately 12% of hematological malignancies [192]. It was noticed that Pirh2 expression is reduced in bortezomib-resistant MM cell lines [190,193] (Table 2). Indeed, Pirh2 overexpression overcame bortezomib resistance and restored the sensitivity of myeloma cells to bortezomib, while alternatively, the reduction in Pirh2 levels correlated with bortezomib resistance. Importantly, it was found that Pirh2 decreased the expression of NF-κB p65, pIKBa, and IKKa and therefore suppressed the canonical NF-κB signaling pathway in MM cells. Thus, it was established that the decreased level of Pirh2 not only associated with MM resistance to bortezomib, but also that high Pirh2 expression was observed more often in patients with newly diagnosed MM than in patients with relapsed MM [190].

## 6. The Role of Pirh2 in Viral Infections

Being a negative regulator of p53, Pirh2 may also affect p53 functions during the viral infection. p53 is a known suppressor of HIV-1 (human immunodeficiency virus 1) expression. p53 affects the HIV-1 infection at multiple levels. For example, in response to IFN-α/β induction, p53 inhibits the transactivation activity of HIV-1 Tat protein [194] required for HIV-1 replication. Furthermore, p53 was repeatedly shown to suppress the transcription from HIF-1 LTR (long terminal repeat) promoter [195,196]. Particularly, Mukerjee et al. demonstrated that p53 prevented phosphorylation of RNA polymerase II, thereby inhibiting viral transcriptional elongation of the virus in HIV-infected cells. Based on this observation, it can be hypothesized that Pirh2 is involved in p53 downregulation during the HIV-1 infection [197]. This assumption was later confirmed by the fact that Pirh2 level is upregulated upon HIV-1 infection in human monocytes cell line, U-937. Pirh2 stabilization is important for p53 downregulation and survival of HIV-1-infected cells. In addition, Shi et al. demonstrated that p53-induced p21 expression blocked HIV-1 reverse transcription and this effect could be rescued by p53 knock-down [198]. Being a negative regulator of p53, Pirh2 may potentially attenuate the p53-mediated antiretroviral mechanism of cell defense. Given the data described above, Pirh2 can be considered as a potential target for anti-HIV drug development [20].

Another RNA-containing and highly pathogenic virus SARS-CoV (severe acute respiratory syndrome coronavirus) was reported to be antagonized by p53, whereby the latter inhibited SARS-CoV replication. In this respect, it was shown that Pirh2 interacts with the papain-like protease of SARS-CoV which leads to stabilization of Pirh2 and subsequent p53 degradation [199].

Prototype foamy virus (PFV) is a retrovirus and member of the foamy viruses family which are spread in many species. Originally, these viruses were isolated from a nasopharyngeal carcinoma patient [200]. In contrast to the well-established virus-promoting role of the p53–Pirh2 axis, Dong et al. reported that Pirh2 acts as an inhibitor of PFV by suppressing its replication, thereby contributing to the latency of infection [201]. Thus, it may be suggested that Pirh2 may too act as an antagonist of viral replication.

Porcine circovirus (PCV) belongs to the viral family Circoviridae and is widespread in the porcine population. To date, two genotypes of PCV are recognized: PCV1 and PCV2. PCV2 infection is associated with lymphohistiocytic infiltrations into granulomatous lymphadenitis, hepatitis, interstitial pneumonia, nephritis, enteritis, myocarditis, and pancreatitis [202]. The PCV2 genome encodes the ORF3 protein which induces aberrant apoptosis of lymphocytes and thus may be responsible for immunosuppression (Shibahara 2000). Mechanistically, ORF3, by binding to Pirh2, sequesters it away from p53, resulting in stabilization of the latter and activation of apoptosis [203]. Moreover, it was demonstrated that the Pirh2–ORF3 interaction results in Pirh2 re-localization to the nucleus. Importantly, nuclear Pirh2 was unable to downregulate p53 in the presence of ORF3 [204]. Thus, Pirh2 activity is suppressed upon PCV infection leading to p53-induced lymphocyte apoptosis.

Measles virus (MV) also known as rubeola virus is responsible for an acute childhood and young adulthood disease. MV is a negative-strand RNA virus whose genome contains six transcriptional units encoding the nucleoprotein, phosphoprotein, matrix protein, fusion protein, hemagglutinin protein, and large protein [205]. It was shown that phosphoprotein (P protein) of the MV interacts with Pirh2 and alters its localization which leads to stabilization of Pirh2 [206]. Specifically, the P protein of MV and Pirh2 formed aggregates in the cytoplasm of HeLa cells and Pirh2-mediated ubiquitination was reduced in the presence of P protein [206]. However, the functional role of Pirh2 in MV is still unclear and needs further investigations.

## 7. The Role of Pirh2 in Regeneration of Nerve Tissue Damage

In addition to participating in progression of different cancer types and viral infections, Pirh2 is also involved in regeneration of the nerve tissue damage and embryonal formation of neuroepithelium.

An inverse correlation was shown between changes in Pirh2 and p27 levels after sciatic nerve injury in rats [207]. In intact nerves, expression of p27 is relatively high. However, after the nerve damage the level of p27 decreases concomitantly with upregulation of Pirh2. This is consistent with the ability of Pirh2 to suppress p27 by ubiquitin-dependent degradation in different cell types including the nerve cells [208]. After the injury, expression of proliferation markers PCNA and Ki67 increased dramatically. The enhanced proliferation of Schwann cells was shown to be associated with Pirh2 stabilization [207]. Schwann cells are known to play a key role in peripheral nerve regeneration, being able to re-enter the cell cycle after damage of the peripheral nerves [209].

After injuring the right cerebral cortex of rats, a significant activation of SCYL1-BP1 was observed. Importantly, SCYL1-BP1 colocalized with Pirh2 in neurons of the injured cerebral cortex [210]. As was discussed above, SCYL1-BP1 is a target of Pirh2-induced ubiquitination and degradation [178]. Since SCYL1-BP1 promotes autoubiquitination of MDM2 and leads to stabilization of the p53 protein [177], Pirh2 may contribute to p53 upregulation in neuronal cells. In turn, the determining role of p53 in the regulation of growth and regeneration of neurites in vitro and in vivo is well known [211]. Additionally, SCYL1-BP1 overexpression inhibits NGF-mediated neurite outgrowth and neuronal regeneration after facial nerve axotomy and affects cortical neuron morphogenesis by activating the SCYL1BP1-Mdm2-p53 axis [211]. Further evidence of Pirh2 participation in nerve tissue development and regeneration was reported in the study demonstrating elevated expression of Pirh2 in the neuroepithelium of the mouse embryo [22]. Thus, Pirh2 apparently contributes to nerve tissue regeneration via different ways, but the role of Pirh2 in human diseases associated with nerve tissue damage needs further investigation.

## 8. Concluding Remarks

In this review, we discussed the role of Pirh2 in different processes and diseases, including several types of cancer and viral infections. According to the multiple studies showing the oncogenic and virus-promoting role of Pirh2, Pirh2 should be considered as a promising target for anticancer and antiviral therapies. Surprisingly, no small molecule inhibitors for Pirh2 have been developed yet. This is in starling contrast with another p53-related E3 ligase, Mdm2. There is a number of small-molecule inhibitors for Mdm2 (for review, see [5]). Our studies have also contributed to the ever-growing list of the p53-Mdm2 blockers [79,212]. Importantly, none of them have been approved for clinic due to the short-lasting effect and side effects. In this respect, it should be noted that Pirh2 knockout mice are viable in contrast to Mdm2 knockouts. This fact allows us to suggest that Pirh2 inhibitors may have less side effects compared to the Mdm2 ones, since the former should be less deleterious for cells with wild type p53. To date, there are only a few reports about Pirh2 inhibitors [213], and we believe that this avenue of research will expand in the near future.

## Figures and Tables

**Figure 1 cells-11-01515-f001:**
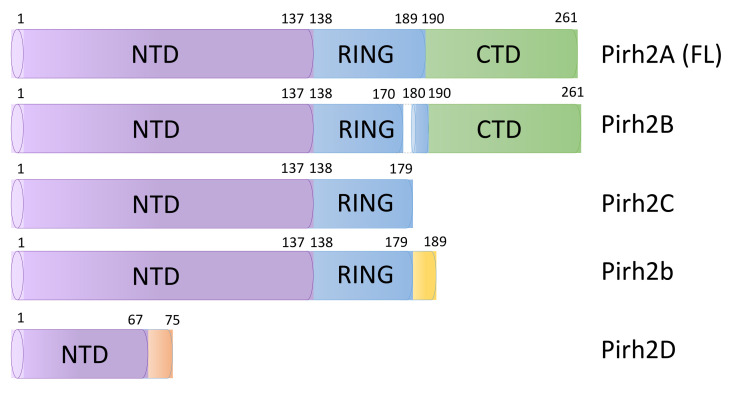
The schematic illustration of Pirh2 domain structure and its isoforms. FL—full-length isoform; NTD—amino-terminal domain; RING—RING domain; CTD—carboxy-terminal domain.

**Figure 2 cells-11-01515-f002:**
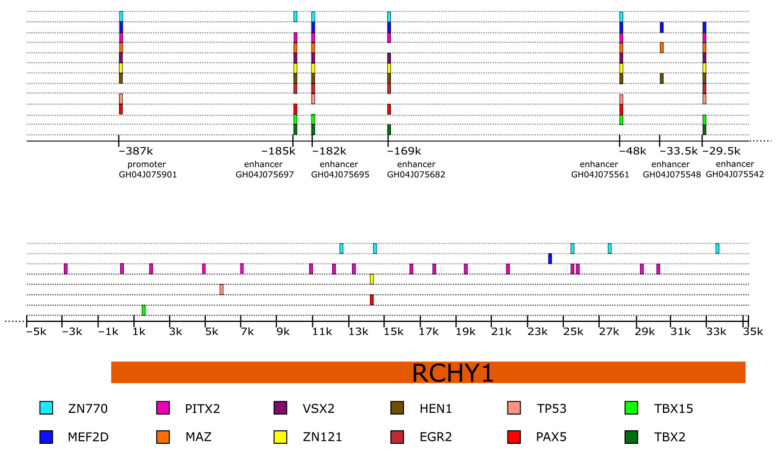
The scheme of potential binding sites of transcription factors in the Pirh2-coding gene and its regulatory elements. k—1000 base pairs relative to the transcription start site.

**Table 1 cells-11-01515-t001:** Pirh2 regulating proteins.

Protein	Type of Regulation	Protein Type	Effect	Reference
p53	Transcriptional	Transcription factor	Activates Pirh2 expression in mouse models. The p53-binding site was detected in intron 3 of RCHY1 mouse gene.	[15]
p63	Transcriptional	Transcription factor	TA isoform of p63 activates Pirh2 expression	[17]
RelA/p65 subunit of NFkB	Transcriptional	Transcription factor	Activates Pirh2 expression binding to the RCHY1 promoter region	[18]
CamKII	Post-translational	Kinase	Phosphorylates Pirh2 at T154 and S155, enhances Pirh2 auto-ubiquitination	[19]
Cdk9	Post-translational	Kinase	Phosphorylates Pirh2 at S211 and T217, enhances Pirh2 auto-ubiquitination	[20]
Hoxa2	Post-translational	Transcription factor	Physically interacts with Pirh2 and 20S proteasome subunits and promotes proteasomal degradation of Pirh2 in a ubiquitin-independent manner	[21]
				[22]
Tip60	Post-translational	Acetyltransferase	Interacts with and stabilizes Pirh2	[23]
PLAGL2	Post-translational	Transcription factor	Physically interacts with Pirh2 dimers and this interaction stabilizes Pirh2 protein level	[24]
RNF144B	Post-translational	Ubiquitin ligase	Ubiquitinates Pirh2 and targets it for degradation through ubiquitin-proteasome system	[25]

**Table 2 cells-11-01515-t002:** The role of Pirh2 in different cancer types.

Cancer Type	Oncogene or Tumor Suppressor	The Effect of Pirh2	Reference
Lung cancer	Oncogene	Pirh2 level is upregulated in NSCLC samples, promotes p53 degradation	[16,179]
	Oncogene	Pirh2 level is upregulated in NSCLC samples, increases cell proliferation	[180,181]
	Oncogene	Increases tumorigenic of NSCLC cells, promotes c-Myc expression	[70,103]
	Oncogene	Increases autophagy level and doxorubicin resistance of NSCLC cells	[182]
Prostate cancer	Oncogene	Negative prognostic factor, enhances ER-mediated PSA expression and suppresses HDAC1	[167]
Oral cancer	Oncogene	Negative prognostic factor, activates OSCC cell migration	[183]
Hepatocellular carcinoma	Oncogene	Negative prognostic factor, Pirh2 level is upregulated in HCC tissues	[184]
	Oncogene	Negative prognostic factor, downregulates p27 in HCC cells	[185]
Head and neck cancer	Oncogene	Negative prognostic factor, promotes proliferation of HNSCC cells	[186]
Glioma	Oncogene	Negative prognostic factor, promotes proliferation and suppresses apoptosis in human glioma cells U87MG	[187]
	Tumor suppressor	Induces apoptosis level in human glioma cells U251 in response to irradiation	[188]
Breast cancer	Oncogene	Negative prognostic factor, Pirh2 level is upregulated in BC samples, promotes proliferation and stabilizes β-catenin level in BC cells	[189]
Multiple myeloma	Tumor suppressor	Pirh2 reduced in bortezomib-resistant MM cells. Pirh2 downregulates NF-κB signaling pathway in MM. Overexpression of Pirh2 overcomes bortezomib resistance.	[190]

NSCLC—non-small cell lung carcinoma; OSCC—oral squamous cell carcinoma; HCC—hepatocellular carcinoma; HNSCC—head and neck squamous cell carcinoma; BC—breast cancer; MM—multiple myeloma.

## Data Availability

Not applicable.

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
