# Peer review of "The Role of E3 Ligase Pirh2 in Disease"

_cells, 2022, doi:10.3390/cells11091515_

Round 1

Reviewer 1 Report

The review « The role of E3 ligase Pirh2 in disease » by Daks et al. builds on a vast experience of Nick Barlev’s lab on the subject. The review is well written and comprehensive.  I recommend publication after a minor revision.

Minor remarks

  1. Figure 2. The authors have identified potential regulators of Pirh2 expression. It would be interesting to compare their expression profile in different tissues with that of Pirh2 to strengthen their analysis
  2. Line 175. The authors mention that Pirh2 is relocalized to the nucleolus in cells overexpressing TIP60. Daks et al. have recently shown that Pirh2 also interacts with another nucleolar component, HuR. Can the authors speculate of a significance of this relocalization and a potential role (of any) of Pirh2 in relation to the nucleolus?
  3. Table 2. Are there any reports linking Pirh2 to lymphomas/leukemias?
  4. Pirh2 is located on 4p21. Deletions in this region are associated with intellectual disability (PMID: 27604828). Any role for Pirh2 in this syndrome?

Author Response

  1. We absolutely agree with the Reviewer that expression profiles of Pirh2 and potential Pirh2-regulating transcription factors in different tissues should have been analyzed. In fact, currently we are performing the bioinformatical experiments to validate in silico potential Pirh2-specific TFs. Once these data are obtained, we plan to publish these results as an experimental paper. In addition, we are planning to include the analysis of Pirh2 expression along with the expression of its potential regulatory TFs in different tissues. As an example, we performed a correlative analysis of top 7 tissues expressing Pirh2 most abundantly and one of the potential Pirh2-regulators, MEF2D, and revealed that 5 of 7 tissue types overlapped for Pirh2 and MEF2D (Figure 1).
  2. We appreciate the Reviewer for this comment and added the corresponding part to the main text (lines 339-345).
  3. We did our best to summarize all reports demonstrating the role of Pirh2 in different types of malignancies and could not find any papers reporting on the role of Pirh2 in human lymphomas or leukemias. This hypothesis would be very interesting to test.
  4. We thank the Reviewer for the very interesting connection. We’ve searched the information regarding the role of Pirh2 in intellectual disability syndrome, but as of today, there is no published data on the Pirh2 role in this disease.

Reviewer 2 Report

The manuscript by Daks etal systemically reviews the molecular characterization and cellular function of E3 ligase Pirh2, including functional domains, isoforms, substrates of Pirh2-mediated ubquibitination, and regulatory mechanisms of Pirh2.  They further introduced association of Pirh2 with human diseases, especially cancers, and discussed the potential clinical application by targeting Pirh2.  The authors also gave their perspective on Pirh2 research. Overall, the manuscript is logically structured and well written, and it covers important findings of the research of Pirh2. A minor concern is that Ref. 70 has been retracted by the journal, and the authors may need to pay attention to it.

Author Response

We thank the Reviewer for careful reading our manuscript. We decided to delete the mentioned reference from the text.